# Highly nitrogen doped carbon nanofibers with superior rate capability and cyclability for potassium ion batteries

Yang Xu[1], Chenglin Zhang[2], Min Zhou[1], Qun Fu[2], Chengxi Zhao[3], Minghong Wu [2] & Yong Lei [1]

Potassium-ion batteries are a promising alternative to lithium-ion batteries. However, it is challenging to achieve fast charging/discharging and long cycle life with the current electrode materials because of the sluggish potassiation kinetics. Here we report a soft carbon anode, namely highly nitrogen-doped carbon nanofibers, with superior rate capability and cyclability. The anode delivers reversible capacities of 248 mAh g$^{-1}$ at 25 mA g$^{-1}$ and 101 mAh g$^{-1}$ at 20 A g$^{-1}$, and retains 146 mAh g$^{-1}$ at 2 A g$^{-1}$ after 4000 cycles. Surface-dominated K-storage is verified by quantitative kinetics analysis and theoretical investigation. A full cell coupling the anode and Prussian blue cathode delivers a reversible capacity of 195 mAh g$^{-1}$ at 0.2 A g$^{-1}$. Considering the cost-effectiveness and material sustainability, our work may shed some light on searching for K-storage materials with high performance.

---

[1] Institut für Physik & IMN MacroNano (ZIK), Technische Universität Ilmenau, Ilmenau 98693, Germany. [2] School of Environmental and Chemical Engineering, Shanghai University, Shanghai 200444, China. [3] Department of Chemistry, Key Laboratory for Advanced Materials and School of Chemistry & Molecular Engineering, East China University of Science and Technology, Shanghai 200237, China. Correspondence and requests for materials should be addressed to M.W. (email: mhwu@shu.edu.cn) or to Y.L. (email: yong.lei@tu-ilmenau.de)

Lithium-ion batteries (LIBs) are the dominating power sources of portable electronics, and are considered as potential technology for electric vehicles, renewable energy storage, and smart grids[1]. The mass production of LIBs raises concerns about the heavy reliance on them due to the rising costs and availability of global lithium resources[2]. One basic requirement for stationary batteries is low-cost upon scaling up, where economies of scale should be applicable. This motivates the pursuit of alternative batteries based on earth-abundant elements[3,4]. Sodium-ion batteries (SIBs) have received a resurgence of attention, owing to the abundance of sodium and similar electrochemistry with LIBs. Significant advancement for SIBs has been achieved in the last few years, demonstrating encouraging capacity, cycle life, and rate capability[5–9]. Recently, the potassium-ion batteries (PIBs) came into spotlight because potassium has comparable abundance as sodium, and PIBs could offer a higher energy density, given the lower redox potential of potassium than sodium (–2.92 V vs. –2.71 V). In contrast to SIBs, only limited amount of studies on PIBs have been reported[10–12].

Graphite is a standard anode in commercial LIBs, but cannot be directly implemented to SIBs in the presence of carbonate electrolyte[13], which is a major barrier toward the commercialization of SIBs. It was demonstrated recently that only solvated Na ions could be intercalated into graphite with appropriate electrolytes[13–15]. Naturally, the failure of intercalating the desolvated Na ions into graphite has intrigued analogous investigation in PIBs[16–18]. Jian et al.[16] and Luo et al.[17] demonstrated that electrochemical procedure can be used to achieve $KC_8$ (stage 1) potassiation and reversible depotassiaion in graphite, generating a capacity of ~250 mAh g$^{-1}$ at a low rate (≤C/10, 1C = 279 mA g$^{-1}$). Other graphitic materials such as reduced graphene oxide[17] and polynanocrystalline graphite[19] have been examined for K-storage. Graphitic carbon possesses an ordered structure with confined space to store the K ions, which causes a detrimental impact on the battery performance, considering the large size of K-ion. As a result, graphitic carbon often exhibits moderate cycle life and unfavorable rate capability.

Nongraphitic soft carbon appears to be amenable in K storage because of the disordered structure. Soft carbon as PIB anodes was first reported by Ji's group[16]. The carbon exhibited a comparable low rate capacity and better cyclability than graphite. Liu et al. conducted an in situ study of the electrochemically driven potassiation of an individual carbon nanofiber consisting of a bilayer wall with an outer layer of nongraphitic soft carbon enclosing an inner layer of graphitic carbon[20]. The study showed that the soft carbon layer exhibited about three times more volume expansion than the graphitic carbon layer after complete potassiation, suggesting a higher K-storage capacity. So far, research on soft carbons has been very limited[16,20,21], and the reported cycle life and rate capability are rather moderate. There is room to push forward the performance of soft carbon and realize both great rate capability and cyclability.

Soft carbon often displays a sloping charge/discharge profile that is related to ion storage dominated by the process occurring on the surface defects. Increasing surface defects could enable great rate capability and cyclability based on two reasons. First, the surface-dominated K-storage takes place mainly on the surface and near-surface region, and can proceed with fast kinetics, making it favorable for high rate capability. Second, surface-dominated K-storage can well maintain the structure of the carbon because of the reduced K-intercalation, making it favorable for enhancing the cyclability. In this regard, introducing nitrogen atoms in carbon has shown attractiveness to promote the surface-dominated charge storage[22,23]. N-doping can produce defects in carbons and increase the active sites for ion storage[24,25]. The N-doped carbons have higher electronegativity than undoped

counterparts, forming a stronger interaction between ions[26,27]. Recently, Share et al. demonstrated that N-doping of the graphitic carbon (few-layered graphene with a doping level of 2.2 at%) could largely increase the K-storage capacity in graphite[28]. The N-doped hard carbon microspheres reported by Chen et al. exhibited high-rate capability, enabled by the surface-driven K-storage[29]. The correlation of N-doping species to the K-storage performance was not presented due to the natural difficulty to control the doping level when dealing with hard carbons, and a full cell demonstration was missing too. Both aspects are critical to utilize N-doped carbons to the maximum extent. Given the advantages of soft carbon over graphitic and hard carbon in PIBs[16], it is of high importance to study the N-doped soft carbon and the correlation of N-doping species to K-storage performance.

In this work, we demonstrate high-rate capability and great cyclability of K-storage in soft carbon, enabled by a surface-dominated charge storage process. The N-doped carbon nanofibers (NCNFs) with a high N-doping level of 13.8% are synthesized, and exhibit a high reversible capacity (248 mAh g$^{-1}$ at 25 mA g$^{-1}$), excellent rate capability (104 and 101 mAh g$^{-1}$ at 10 and 20 A g$^{-1}$, respectively), and great cyclability (4000 cycles) as anode materials in half cells. To the best of our knowledge, the obtained rate performance and cycle life in K-storage are among the very few best results of all the reported carbon allotropes. Theoretical investigation shows the higher K-adsorption ability and K-ion diffusivity upon pyrrolic and pyridinic nitrogen doping. Furthermore, we present a cost-effective and material sustainable PIB full cell, based on the NCNFs and potassium Prussian blue. The full cell delivers a reversible capacity of 195 mAh g$^{-1}$ that is the highest value for the PIB full cells so far. Its energy density of 130 Wh kg$^{-1}$ is comparable to commercial LIBs. Our results may bring a shift from graphitic carbon to soft carbon in the study of PIBs and offer understandings of material engineering strategies to boost the performance of PIBs.

## Results

**Characterization.** We utilized polypyrrole (PPy) nanofibers as a precursor for direct carbonization because of their high starting N content. The precursor was synthesized by an oxidative template assembly approach[30]. It shows a homogeneous morphology of the cross-linked nanofibers, with a diameter of 70–80 nm (Supplementary Fig. 1). It was carbonized in $N_2$ at temperatures of 650 °C, 950 °C, and 1100 °C for 2 h, with the resultant specimens being labeled NCNF-650, NCNF-950, and NCNF-1100, respectively. As shown in the Fourier transform infrared (FTIR) spectra (Supplementary Fig. 2), the precursor exhibits well-defined peaks that are characteristic of PPy[31]. The spectra of all three NCNFs exhibit broad bands due to the strong absorption of carbon with total elimination of the PPy peaks[32], indicating complete carbonization.

Figure 1a, b shows representative scanning electron microscopy (SEM) images of NCNF-650, and those of NCNF-950 and NCNF-1100 are shown in Supplementary Fig. 3. NCNFs well inherited the cross-linked morphology with the shrinkage of diameter. Figure 1c is a transmission electron microscopy (TEM) image of NCNF-650, which highlights the hollow interior of the nanofibers. The sidewall of the nanofibers is about 20 nm thick, giving an inner diameter of 30–40 nm. The rough surface indicates the structural defects that are favorable for K-storage[33,34]. NCNF-950 and NCNF-1100 show the same structural features (Supplementary Fig. 3). As shown in the high-resolution TEM image (Fig. 1d), NCNF-650 comprises of turbostratic domains that consist of highly curved graphene layers. With increasing carbonization temperature, NCNF-950

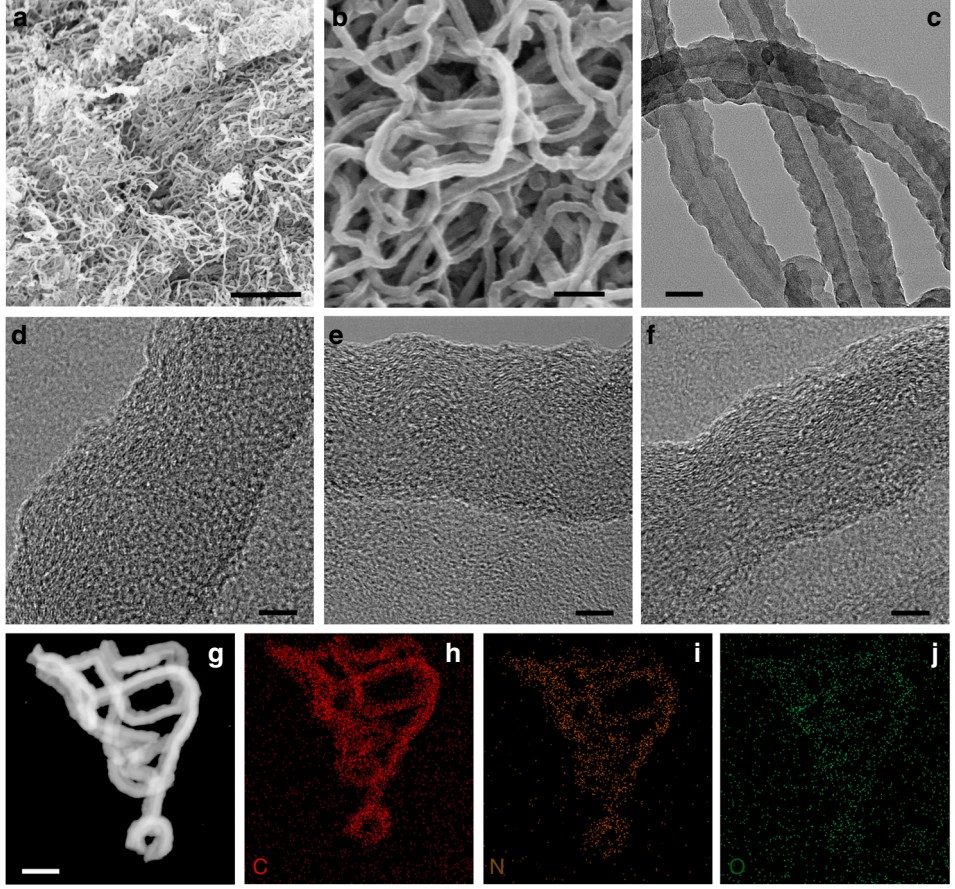

**Fig. 1** Structure characterizations of NCNFs by electron microscopies. **a, b** SEM and **c** TEM images of NCNF-650. HRTEM images of NCNF-650 (**d**), NCNF-950 (**e**), and NCNF-1100 (**f**). **g–j** Images of element mapping of C, N, and O for NCNF-650. Scale bars: 2 μm (**a**); 200 nm (**b**); 50 nm (**c**); 5 nm (**d–f**); 100 nm (**g**)

(Fig. 1e) and NCNF-1100 (Fig. 1f) become progressively ordered with the graphene layers being less curved and more aligned, which reveals the graphitizable nature of the soft carbons. However, none of the specimens shows the presence of equilibrium graphite. Element mapping images (Fig. 1g–j) demonstrate the incorporation of N and its even distribution over the entire nanofiber. Species and content of the N dopants will be discussed later.

The structure of NCNFs was characterized by X-ray diffraction (XRD) and Raman spectroscopy. All the XRD patterns (Fig. 2a) have two broad diffraction peaks near 25° and 43°, which can be indexed to (002) and (100) planes, respectively. Upon increased temperature, the (002) peak shifts to a higher angle, indicating that the graphene interlayer space becomes smaller. However, as will be later shown in the sloping discharge profiles, the absence of a graphite-like low-potential plateau indicates that K-intercalation into graphene layers does not appear to be a major contributor to the overall capacity. We calculated an empirical $R$ value (Supplementary Fig. 4) based on the ratio of the (002) intensity and background at the equivalent peak, according to the study of Dahn et al.[35]. A lower $R$ value suggests a lower degree of graphitization. As listed in Table 1, the $R$ value increases from 2.5 for NCNF-650 to 3.6 and 4.3 for NCNF-950 and NCNF-1100, respectively, once again confirming the graphitizable nature of the carbons. The Raman spectra (Fig. 2b) display two peaks centered at 1353 and 1567 cm$^{-1}$, corresponding to the disorder-induced D-band and in-plane vibrational G-band, respectively. The intensity ratio of the two peaks, termed as $I_G/I_D$, can be used to indicate the degree of graphitization[36]. We calculated the $I_G/I_D$

value using the absolute heights of the peaks in the spectra and the value increases at higher temperature (Table 1), suggesting an increased graphitization degree and being agreed with the observations from TEM and XRD. The overall obtained results are in accordance with the previously reported carbons prepared by carbonization at different temperatures[22,31], which proves that the incorporation of nitrogen does not induce major change of the graphitization degree.

The surface chemistry of NCNFs was investigated by X-ray photoelectron spectroscopy (XPS). The survey spectra (Fig. 2c) show that NCNFs are composed of C, N, and O, with the N content being 13.8%, 8.1%, and 4.9% (at%) for NCNF-650, NCNF-950, and NCNF-1100, respectively. Clearly, the carbonization temperature is a critical factor in determining the N content, and its dependence on temperature agrees with the generally reported trend[37,38]. The N content of NCNF-650 is among the highest reported values for carbon[22,31,39]. Oxygen was detected, as shown in the survey spectra and element mapping (Fig. 1j). It has been argued that surface oxygen-containing functional groups could contribute to the battery performance of carbons[40,41], but the contribution occurs in a voltage range higher than 1.5 V (vs. Na/Na$^+$). Considering the relatively even O content (Table 1) and much higher N content, as well as the voltage window of major capacity contribution (<2 V, vs. K/K$^+$) in our case, it is reasonable to correlate the performance difference among the specimens primarily to the N dopant. Figure 2d shows the high resolution N 1s core level XPS spectra. The N species in the PPy precursor is pyrrolic N (N-5 at ~399.8 eV), which is ascribed to the N atoms within the pentagonal

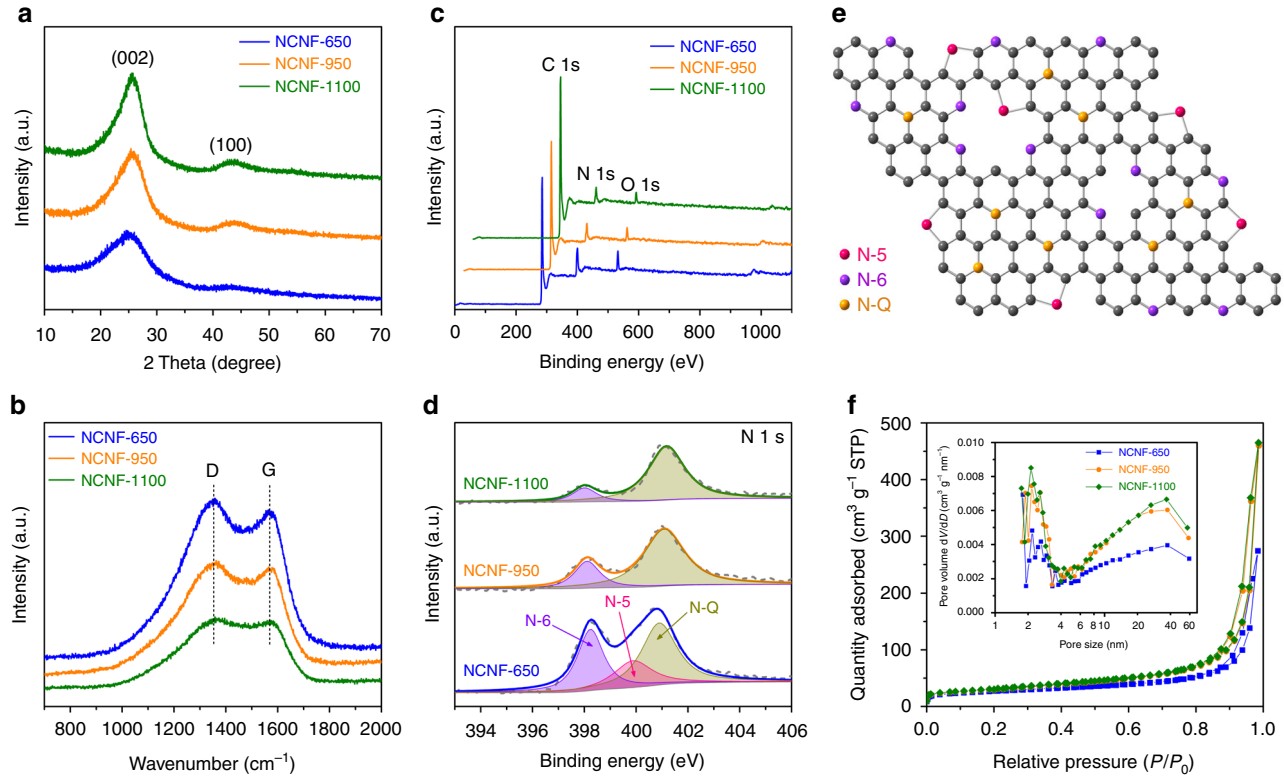

**Fig. 2** Structure characterizations of NCNFs by spectroscopies. **a** XRD patterns. **b** Raman spectra. **c** XPS survey spectra. **d** N 1s core level XPS high-resolution spectra. **e** A schematic illustrating the structure of the N-doping species. **f** Nitrogen adsorption–desorption isotherms (inset shows the related pore size distribution)

**Table 1 Structure properties and surface chemistry of the NCNFs**

| | $R$ | $I_G/I_D$ | $S_{BET}$ (m² g⁻¹) | $V_t$ (cm³ g⁻¹)ᵃ | Element content (at%) | | | % of total N 1s | | | C (mAh g⁻¹)ᵇ |
|---|---|---|---|---|---|---|---|---|---|---|---|
| | | | | | C | N | O | N-6 | N-5 | N-Q | |
| NCNF-650 | 2.5 | 0.94 | 99 | 0.42 | 80.6 | 13.8 | 5.6 | 33.3 | 22.7 | 44.0 | 368 |
| NCNF-950 | 3.6 | 0.96 | 107 | 0.71 | 87.8 | 8.1 | 4.1 | 23.8 | - | 76.2 | 297 |
| NCNF-1100 | 4.3 | 0.98 | 110 | 0.72 | 92.1 | 4.9 | 3.0 | 15.2 | - | 84.8 | 281 |

ᵃ The total pore volume was determined at a relative pressure of 0.98
ᵇ The values are the first charge capacity at current density of 25 mA g⁻¹

pyrrole rings (Supplementary Fig. 5). Two additional species can be found in NCNFs: pyridinic N (N-6 at ~398.2 eV) and quaternary N (N-Q at ~410.0 eV). NCNF-650 contains all three species, while NCNF-950 and NCNF-1100 contain only N-6 and N-Q (Table 1). Such N distribution qualitatively agrees with reports on pyrolyzed pyrrole[22,37]. The content of N-6 relative to N-Q reduces with increasing temperature, indicating that higher temperature promotes the generation of N-Q. The high-resolution C 1s core level XPS spectra (Supplementary Fig. 6) provides a supplementary proof of the reduced N content and increased degree of graphitization upon increasing temperature. The structure of the N species is illustrated in Fig. 2e. N-5 and N-6 should be highly chemically active, and are likely to create additional defects in the graphene layers. Such moieties and associated defects are expected to enhance the capacity by reversibly binding with the charge carriers and exhibiting fast kinetics, as compared with the more inert N-Q[24,26,38,39].

The surface area and porous texture of NCNFs were analyzed using N₂-adsorption. Figure 2f shows the nitrogen adsorption–desorption isotherms, and the inset highlights the

corresponding pore size distribution. All NCNFs exhibit type IV isotherms with type H3 hysteresis loops, showing a meso/macroporous structure. The Brunauer–Emmett–Teller (BET) surface areas were calculated to be 96, 107, and 110 m² g⁻¹ for NCNF-650, NCNF-950, and NCNF-1100, respectively. This indicates that carbonization temperature has minimal effect on the resultant surface area in the employed range. Pores that are smaller than 4 nm are generated from mass loss during carbonization, and those larger than 20 nm are ascribed to the hollow interior of the fibers. The mesopores could offer fast ion adsorption and short diffusion distance, contributing to fast charging/discharging properties of the NCNFs.

**Electrochemical performance as PIB anodes in half cells**. The K-storage behavior of NCNFs was first tested using cyclic voltammetry (CV) in a voltage window of 0.01–3.0 V (vs. K/K⁺). As shown in Fig. 3a and Supplementary Fig. 7, during the first cathodic scan, a peak appears at ~0.55 V and disappears in the second scan, which is ascribed to the formation of solid

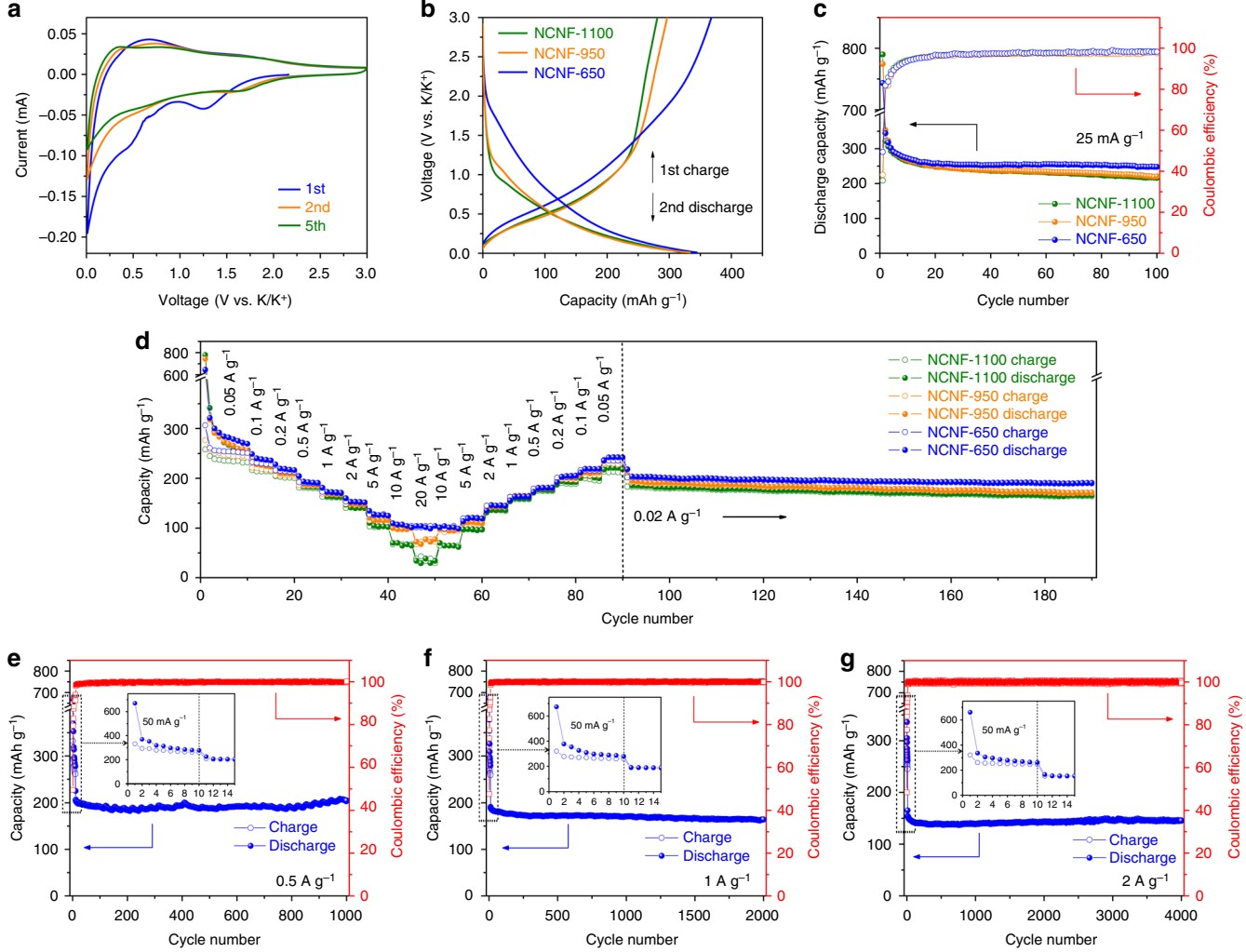

**Fig. 3** Electrochemical performance of NCNFs as PIB anodes in half cells. **a** CV curves of NCNF-650. **b** First charge and second discharge profiles of NCNFs. **c** Cycling performance of NCNFs at a current density of 25 mA g$^{-1}$. **d** Rate performance of NCNFs with rates ranging from 0.05 to 20 A g$^{-1}$. Long-term cycling performance of NCNF-650 at high rates of 0.5 (**e**), 1 (**f**), and 2 A g$^{-1}$ (**g**)

electrolyte interface (SEI)[42,43]. A slope extending to the cutoff voltage can be attributed to the K-intercalation in Super P at low-potential range, which was also observed in our previous work[44]. The first anodic scan features a broad peak over a wide voltage window and a large capacity loss is observed, signaling a low first Coulombic efficiency (CE). During the following cycles, several humps appearing at 1.60 V (NCNF-650), 0.80 V (NCNF-950), and 0.70 V (NCNF-1100) are attributed to the interaction of K ions with N species and may reflect various binding energy[23,45–47]. Increasing carbonization temperature leads to a progressive reduction in the total area of the scan curves, as shown by the loss of the peak at higher voltage range in NCNF-950 and NCNF-1100 (Supplementary Fig. 7). Since higher temperature eliminates the content of N species, the importance of N-doping in determining the total reversible capacity is evidenced.

Supplementary Fig. 8 shows the galvanostatic charge/discharge profiles of the NCNF electrodes at 25 mA g$^{-1}$. A quasi-plateau located between 0.65 and 0.85 V and a long tail extending to 0.01 V can be observed in the first discharge, being in accordance with the CV results. Since Super P maintained a low reversible capacity (Supplementary Fig. 9), its contribution to the overall capacity is expected to be very small, considering its low content (10 wt%) in the electrodes. The first discharge and charge capacities are 744 and 368 mAh g$^{-1}$ for NCNF-650, 775, and 297 mAh g$^{-1}$ for

NCNF-950, and 790 and 281 mAh g$^{-1}$ for NCNF-1100, giving low first cycle CEs of 49%, 38%, and 36%, respectively. Similar trend was observed in the reported carbons in LIBs and SIBs[31,48,49]. Although NCNF-650 has the highest K-adsorption energy and the resultant carbon defects (will be shown later), it shows the least K-intercalating capacity below 0.25 V[16–18], owing to the largest average graphene interlayer spacing (Fig. 2a), which indicates the fewest irreversible K-intercalation[50]. Additionally, it has been reported that doping of nitrogen can, to some extent, suppress the electrolyte decomposition and SEI formation[51,52], which also contributes to a higher first cycle CE of NCNF-650. The CE reaches 90% at cycle 6 for all the NCNF electrodes. It is worth noting that NCNF-650 exhibited a much higher first charge capacity (368 mAh g$^{-1}$) than NCNF-950 and NCNF-1100 (297 and 281 mAh g$^{-1}$, respectively, Fig. 3b), and the capacity is well beyond the theoretical capacity (279 mAh g$^{-1}$) of stage 1 KC$_8$. From the second cycle, the charge/discharge profiles of all NCNFs exhibit a characteristic sloping feature, which is similar to those previously reported in soft carbons[16,46,53,54]. NCNF-650 delivered a reversible capacity of 248 mAh g$^{-1}$ after 100 cycles (Fig. 3c), being among the highest values of carbon materials in PIBs[16,18,21,28,29,55]. NCNF-950 and NCNF-1100 delivered lower capacities of 221 and 216 mAh g$^{-1}$, respectively. Figure 3d shows

the rate capability at various current densities, where NCNF-650 exhibited the best performance. The charge/discharge profiles of NCNF-650 can be found in Supplementary Fig. 10. It delivered capacities of 238, 217, 192, 172, 153, and 126 mAh g$^{-1}$ at 0.1, 0.2, 0.5, 1, 2, and 5 A g$^{-1}$, respectively. Even at current densities as high as 10 and 20 A g$^{-1}$, there are still 104 and 101 mAh g$^{-1}$ retained. Moreover, it maintained good cyclability during the next 100 cycles at 0.2 A g$^{-1}$, delivering a stable capacity of 191 mAh g$^{-1}$. NCNF-950 and NCNF-1100 delivered lower capacities than NCNF-650 at various rates, and the trend follows the order of their N-doping level. The capacity discrepancy becomes more significant with increasing current density, suggesting a critical role of N-doping at high rates. Additionally, we tested the long-term cyclability of NCNF-650 at high rates (Fig. 3e–g). Prior to high rates, NCNF-650 was cycled at 50 mA g$^{-1}$ for 10 cycles. As expected, it exhibited impressive cycling stability by retaining capacities of 205 (0.5 A g$^{-1}$), 164 (1 A g$^{-1}$), and 146 mAh g$^{-1}$ (2 A g$^{-1}$) after 1000, 2000, and 4000 cycles, respectively. TEM measurement was performed on NCNF-650 after 4000 cycles (Supplementary Fig. 11). The nanofiber structure is well maintained, except for the less discernible hollow interior due to the SEI layer. HRTEM image shows the disordered structure and turbostratic domains. Element mapping shows even distribution of N and K over the fibers, indicating the stability of the N dopant and the effective K-storage. Therefore, the great structural stability is evident. The long-term operation of the electrode without much degradation can be attributed to the charge storage action that is mainly on or near the surface of the nanofibers at high rates. Cycling at low rates would give more charge storage contribution in bulk and certainly do more damage to the electrode, as can be seen that NCNF-650 showed lower capacity retention after 100 cycles at 25 mA g$^{-1}$ than at high rates.

**Quantitative analysis of surface-dominated charge storage.** We next investigated the kinetics of the electrodes using CV measurement at scan rates of 0.2 to 10 mV s$^{-1}$ (Fig. 4a–c). In the case of NCNF-650, broad (de)potassiation peaks are maintained at high scan rates and the peak coverage toward high potential can

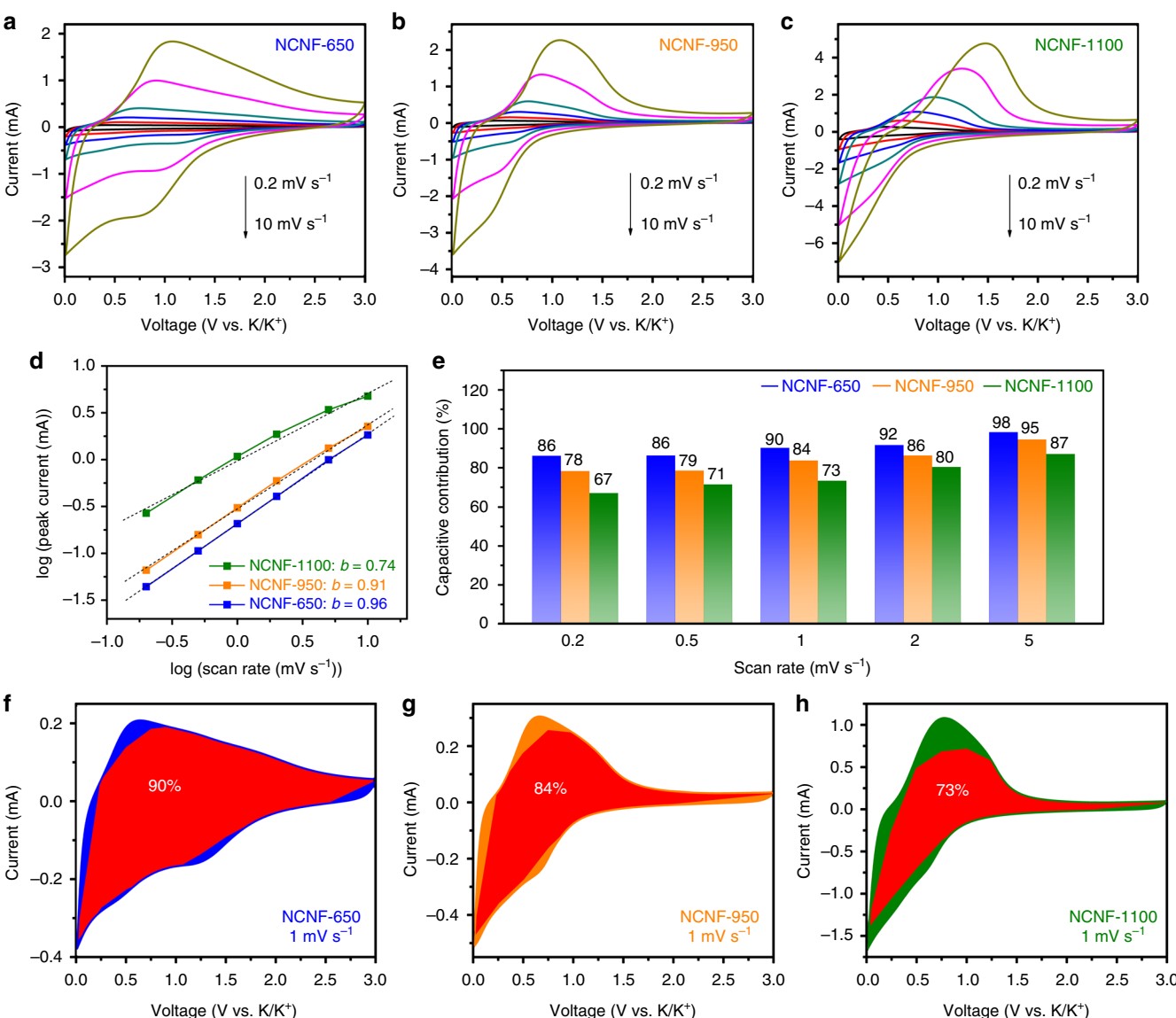

**Fig. 4** Quantitative analysis of surface-dominated K-storage in NCNFs. CV curves of NCNF-650 (**a**), NCNF-950 (**b**), and NCNF-1100 (**c**) at various scan rates of 0.2 to 10 mV s$^{-1}$. **d** *b* value determination. **e** Contribution of the surface process in the NCNFs at different scan rates. Contribution of the surface process at scan rate of 1 mV s$^{-1}$ in NCNF-650 (**f**), NCNF-950 (**g**), and NCNF-1100 (**h**)

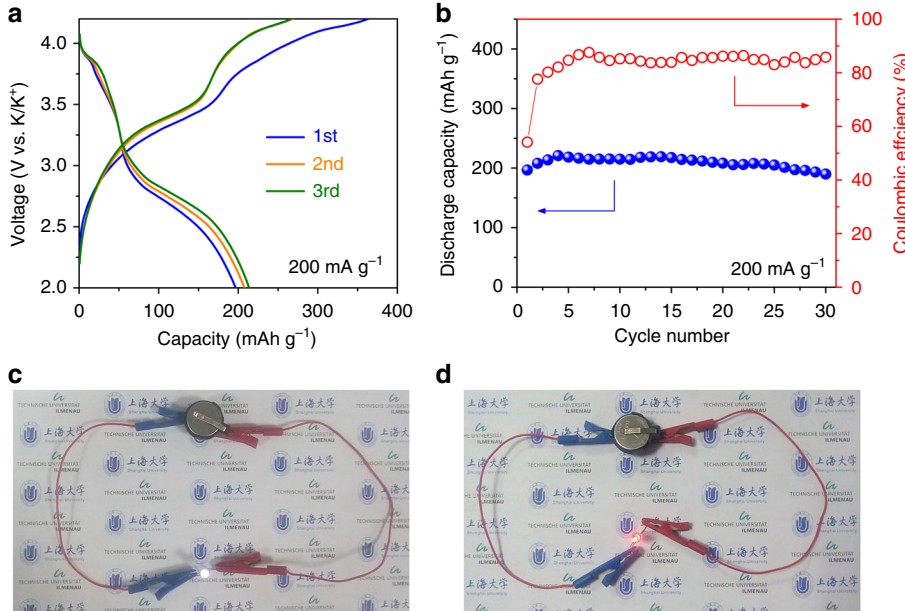

**Fig. 5** Electrochemical performances of the NCNF-650/KPB full cell. Galvanostatic charge/discharge profiles (**a**) and cycling performance (**b**) at a current density of 200 mA g$^{-1}$. Optical photographs of a white LED (**c**) and a red LED (**d**) lightened by the full cell

be clearly seen, being in contrast to NCNF-950 and NCNF-1100, whose peaks become steeper and show large polarization. These observations imply better preservation of the surface-dominated characteristic enabled by higher N-doping level in NCNF-650. The peak current obeys a power law relationship with the scan rate, according to Eq. (1)[56,57]:

$$i = av^b \qquad (1)$$

where $i$ is peak current, $v$ is scan rate, $a$ and $b$ are adjustable constants. By plotting $\log(i)$ against $\log(v)$, the $b$ value can be extracted from the slope. It would be 0.5 for an ideal faradaic intercalation process controlled by semi-infinite linear diffusion while close to 1 for a surface charge storage process free of diffusion control. Figure 4d shows the plot applied on the depotassiation peak current. A good linear relationship can be seen for NCNF-650 and the $b$ value was calculated to be 0.96, suggesting a fast kinetics dominated by a surface process. As expected, this value decreases to 0.91 and 0.74 for NCNF-950 and NCNF-1100, respectively. In particular, data points collected from NCNF-1100 show the worse linear relationship at high scan rates than at low scan rates, reflecting its much lower capacities than those of NCNF-650 and NCNF-950 at high rates. The surface process contribution can be further quantitatively differentiated by separating current response $i$ at a fixed potential $V$ into a surface-dependent process (proportional to $v$) and a diffusion-controlled process (proportional to $v^{1/2}$) by Eq. (2)[56,57]:

$$i = k_1 v + k_2 v^{1/2} \qquad (2)$$

by determining $k_1$ and $k_2$, we can separate the fraction of the two processes. Figure 4f–h shows the typical CV profiles at a scan rate of 1 mV s$^{-1}$ for the current from surface process (red region) in comparison with the total current. A surface-dominated contribution (90%) is obtained for NCNF-650, being higher than those of NCNF-950 (84%) and NCNF-1100 (73%). Using similar analysis, the fraction enlarges with increasing scan rates (Fig. 4e) for all NCNFs. NCNF-650 possesses the highest fraction at all scan rates, which complies with the order of N-doping level. This is not surprising since more N dopant in carbon could induce

more surface defects and edges of graphene layers that could enhance K-ion adsorption and result in fast kinetics.

**Electrochemical performance of PIB full cells**. We assembled full cells using NCNF-650 as anode and potassium Prussian blue (KPB) as cathode that was synthesized using a modified method according to our previous work[44]. The rationale behind this is that both materials are cost-effective and materially sustainable and, as proven in LIBs, the maturity of the commercialization has always relied on carbon-based anodes. Characterizations of KPB in a half cell are provided in Supplementary Fig. 12. The full cell was assembled in an anode-limited configuration. Figure 5a shows the charge/discharge profiles of the full cell at 0.2 A g$^{-1}$ tested in a voltage window of 2.0–4.2 V. Two semi-plateaus can be seen in the ranges of 3.4–3.9 V and 2.5–3.0 V in the initial cycles, which corresponds to the two plateaus displayed in the KPB half cell (Supplementary Fig. 12c). The lowered voltages and less defined flat contours are due to the sloping discharge profiles of NCNF-650. The full cell delivered a first discharge capacity of 197 mAh g$^{-1}$ (based on the anode mass) and retained 97% of the capacity (190 mAh g$^{-1}$) after 30 cycles (Fig. 5b). As far as we know, the presented reversible capacity is the highest among all the reported PIB full cells[44,58–60]. The cell is able to lighten a white (working voltage 2.9–3.3 V, Fig. 5c) and red light-emitting diode (working voltage 1.9–2.2 V, Fig. 5d) after being fully charged. We also tested full cells in a cathode-limited configuration (Supplementary Fig. 13). Similarly, the cells displayed two semi-plateaus and a capacity of 74 mAh g$^{-1}$ after 50 cycles (91% of the first discharge capacity) at 0.1 A g$^{-1}$. CE is lower in the anode-limited configuration than in the cathode-limited configuration. We speculate the reason to be the relatively low CE in the KPB half cell (Supplementary Fig. 12c) and the higher amount of KPB used in the anode-limited configuration given the same amount of NCNF-650[43]. Nevertheless, the results here demonstrate the applicability of N-doped soft carbons in the PIB full cell application.

## Discussion

The results presented above show K-storage in the highly nitrogen-doped hollow NCNFs with both high rate capability and

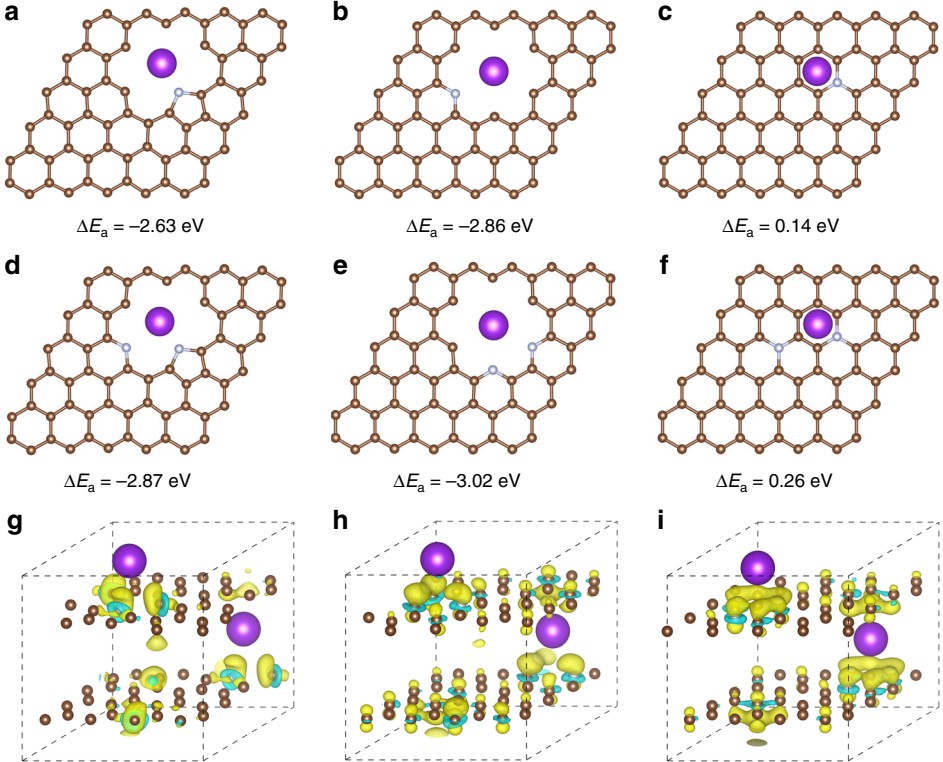

**Fig. 6** Theoretical simulations of K-adsorption in different N-doped structures. Top view of a single K atom adsorbed in the N-5 (**a**), N-6 (**b**), and N-Q (**c**) structures and the corresponding adsorption energy. Top view of a single K atom adsorbed in the doped structures with two nitrogen atoms and the corresponding adsorption energy: **d** one N-5 and one N-6 atoms; **e** two N-6 atoms; **f** two N-Q atoms. Electron density differences of K absorbed in the N-5 (**g**), N-6 (**h**), and N-Q (**i**) structures. Yellow and blue areas represent increased and decreased electron density, respectively. The isosurfaces are the 0.0015 electron bohr$^3$. Brown, silver, and purple balls represent carbon, nitrogen, and potassium atoms, respectively

great cyclability rendered by a surface-dominated process. To further interpret the experimental results, we performed a theoretical investigation of K-adsorption ability of the N-doped structures using first-principles calculations, based on density functional theory (DFT). The three N-doped models, i.e., N-5, N-6, and N-Q, were employed (Fig. 6a–c). The corresponding undoped and carbon defect models, i.e., C-5, C-6, and pristine C (P-C) were also employed to differentiate the doping and defect effects on the K-adsorption ability (Supplementary Fig. 14). A single K atom was placed at different sites in each model, and the optimized geometry structures were obtained at the center of the hollow "holes" by the calculation of adsorption energy ($\Delta E_a$). First, we calculated $\Delta E_a$ with one doped nitrogen atom. The $\Delta E_a$ of N-5 and N-6 are −2.63 and −2.86 eV, respectively, being much higher than that of N-Q (0.14 eV). This indicates that the pyrrolic and pyridinic N-doping has significantly higher K-adsorption ability than the graphitic N-doping. The $\Delta E_a$ of N-Q is even positive, which might be attributed to the electron-richness of the graphitic doping that shows negative effect on K-adsorption[25,61]. Second, the $\Delta E_a$ of C-5 and C-6 (−2.42 and −2.80 eV, respectively) are comparable to those of their N-doped counterparts. This suggests that carbon defects could contribute to K-adsorption, together with N-doping. In fact, the structures of N-5, N-6, C-5, and C-6, all contain defects that provide an electron deficiency and a tendency to gain electron from K atoms[25,62]. NCNF-650 has the lowest graphitization degree that could induce smaller and more curved domains and create more defects, compared with NCNF-950 and NCNF-1100. Owing to the high pyrrolic and pyridinic N-doping level and low graphitization degree, NCNF-650 exhibited the highest capacity among the three electrodes. Share et al. recently reported the capacity enhancement of the N-

doped few layer graphene and correlated it to the distributed K-ion storage at local N-5 doping sites, as opposed to carbon defect sites[28]. However, our simulation results shown here suggest the capacity enhancement to be related to both N-doping and carbon defects. We speculate that the different nature of carbons, i.e., high graphitic-few layer graphene in their work and low graphitic soft carbon in our work, might be responsible for the different results. Third, $\Delta E_a$ remains considerably high when two doped nitrogen atoms exist in differently relative positions (Fig. 6d–f and Supplementary Fig. 15), which could represent the coexistence of N-5/N-6 or N-6/N-6 at the same defect point. The highest $\Delta E_a$ are −2.87 and −3.02 eV for the N-5/N-6 and N-6/N-6 combinations, respectively, whereas the $\Delta E_a$ of two N-Q atoms doping (0.26 eV) is more positive than that of single N-Q doping. Therefore, the above calculations unambiguously support the benefits of the pyrrolic and pyridinic N-doping. To understand the bonding nature of the adsorbed K atoms, we calculated electron density difference by subtracting the charge densities of K atoms and bare carbon atoms from those of the combined compounds (Fig. 6g–i and Supplementary Fig. 16). In all cases, there is a net gain of electronic charge in the intermediate region between K atoms and the carbon layers, which indicates a charge transfer from the adsorbed K to its nearest-neighboring C atoms, suggesting an ionic character of the bonding[63]. The charge density is transferred to the bonding carbons in P-C, whereas the charge density tends to accumulate more around the N-doping sites in the doped structures. Moreover, the tendency is more significant in N-5 and N-6 than in N-Q. Therefore, the pyrrolic and pyridinic N-doping lead to stronger K-adsorption than the graphitic N-doping, supporting the experimental results.

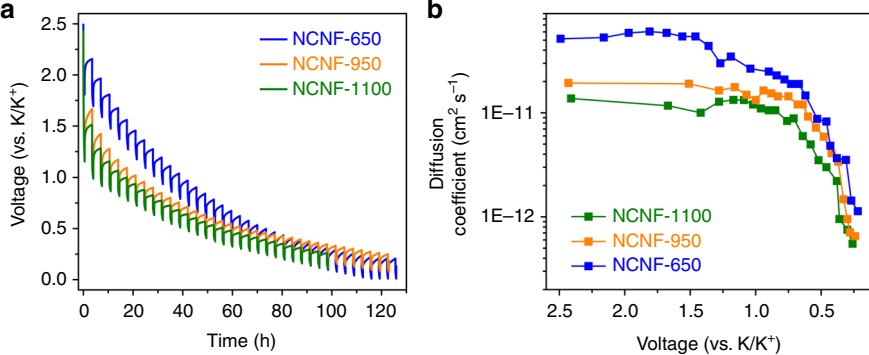

**Fig. 7** Study of the K-ion diffusion coefficient of the NCNF electrodes. **a** GITT profiles of the discharging process. **b** The K-ion diffusion coefficient as a function of the state of discharging process

With the simulation results established, we further seek understanding from a kinetic point of view by using Galvanostatic intermittent titration (GITT) technique to evaluate the K-ion diffusion coefficient ($D_k$) during potassiation. Figure 7a shows the potential response of the electrodes during GITT measurement. The potential change during each relaxation period represents overpotential at the corresponding potassiation stage[61]. In general, NCNF-650 exhibited the smallest over-potentials, whereas NCNF-1100 exhibited the largest ones. The difference is more obvious in the higher voltage range than the lower range. This implies the better kinetic property of NCNF-650. The calculated values of $D_k$ as a function of potential are shown in Fig. 7b and the calculation can be found in Supplementary Fig. 17. All the three electrodes showed a progressively decreasing $D_k$ with the potassiation proceeding toward 0.5 V, which is associated with the sloping characteristic of the discharge profiles. It is obvious that NCNF-650 exhibited the highest $D_k$, being 1.6–2.9 and 2.1–4.6 times higher than NCNF-950 and NCNF-1100, respectively. It suggests that the K-ion diffusion is much faster in NCNF-650, owing to the easily accessible sites in the carbon structure derived from nitrogen doping, which is in accordance with the higher capacity contribution above 0.5 V. As these sites are progressively potassiated, K ions have to overcome a repulsive charge gradient from the previously bound K ions on the defect sites in order to diffuse further inside the turbostratic domains[64]. This is responsible for the steep drop of $D_k$ below 0.5 V.

In summary, we reported a facile and scalable process to fabricate highly nitrogen-doped soft carbon nanofibers from the polypyrrole precursor. The carbon materials, termed NCNFs, exhibited excellent electrochemical performances as anodes in PIBs, demonstrating superior rate capability and cyclability among all the reported PIB anodes. Quantitative analysis and theoretical simulations were employed to interpret the benefits of nitrogen doping and demonstrate the advantage of the pyrrolic and pyridinic N dopants over the quaternary N dopant. Moreover, full cells based on the NCNFs and potassium Prussian blue delivered the highest reversible K-ion storage capacity so far, further indicating that NCNFs should be considered as a promise candidate of anode materials to realize high performance PIBs. In addition, our work offers a practical pathway to tune K-storage performance through compositional adjustment and opens up the opportunity of using disordered carbon materials in the exciting area of PIBs.

## Methods

**Preparation of PPy precursor**. The PPy nanofibers were synthesized by a modified oxidative template assembly route. In a typical synthesis, cetrimonium bromide (CTAB, ($C_{16}H_{33}$)-$N(CH_3)_3$Br, 0.8 g) was dissolved in hydrochloric acid

solution (HCl, 240 ml, 1 mol L$^{-1}$) under ice bath (0–3 °C) to form a transparent solution. Ammonium persulfate (APS, $(NH_4)_2S_2O_8$, 1.2 g) was then added into the above solution under magnetic stirring, and a white suspension was formed immediately. Afterward, pyrrole monomer (1.6 ml) was dropwise added into the white suspension, and polymerization was carried out for 3 h under stirring. A black precipitate, namely PPy precursor, was obtained, collected by filtration, and washed with deionized water until the filtrate became colorless. The PPy precursor was dried at 80 °C in a vacuum oven for 24 h.

**Preparation of NCNFs**. The NCNFs were obtained by annealing the PPy precursor in N$_2$ atmosphere for 2 h at temperatures of 650, 950, and 1100 °C. The ramp rate was 5 °C min$^{-1}$.

**Preparation of KPB**. KPB was synthesized by a facile room temperature-precipitation method in an aqueous solution. In a typical synthesis, potassium citrate tribasic monohydrate ($K_3C_6H_5O_7·H_2O$, 5 mmol) and iron chloride (FeCl$_2$, 5 mmol) were dissolved in deionized water (50 ml) under magnetic stirring to form solution A. Potassium hexacyanoferrate (II) trihydrate ($K_4Fe(CN)_6·3H_2O$, 5 mmol) was dissolved in deionized water (50 ml) under magnetic stirring to form solution B. Solution B was dropwise added into solution A under stirring, and precipitation occurred immediately. The mixture was stirred for 2 h and aged for another 10 h. The obtained dark blue precipitates were collected by centrifugation, washed with deionized water and ethanol, and dried at 80 °C in a vacuum oven for 24 h.

**Characterizations**. XRD analysis was performed on a 18 KW D/MAX2500V PC diffractometer using Cu Kα ($\lambda = 1.54$ Å) radiation at a scanning rate of 2° min$^{-1}$. SEM analysis was conducted using a Hitachi S4800 field emission scanning microscopy. TEM and EDS analysis was performed on a JEOL JEM-2100F transmission electron microscope. X-ray photoelectron spectra were acquired on a Thermo SCIENTIFIC ESCALAB 250Xi with Al Kα (hυ = 1486.8 eV) as the excitation source. The binding energies obtained in the XPS spectra analysis were corrected for specimen charging by referencing C 1s to 284.8 eV. Raman spectra were recorded at room temperature with an inVia Raman microscope. BET measurement was conducted using a Quantachrome autosorb IQ automated gas sorption analyzer.

**Electrochemical measurements**. Electrodes were fabricated by mixing NCNFs, Super P, and carboxymethyl cellulose (CMC) sodium salt at a weight ratio of 80:10:10, then coated uniformly (doctor-blade) on a copper foil with a mass loading of ~1.5 mg cm$^{-2}$. The electrodes were dried at 110 °C under vacuum for 12 h. Electrochemical tests were carried out using a coin cell configuration, CR2032, which was assembled in an N$_2$-filled glovebox with oxygen and moisture concentrations kept below 0.1 ppm. K foil used as counter electrode was separated from working electrode using a glass microfiber filter (Whatman, Grade GF/B). The electrolyte was 0.8 M KPF$_6$ in ethylene carbonate and propylene carbonate (EC:PC = 1:1). CV was performed on a VSP electrochemical workstation (Bio-Logic, France). Galvanostatic charge/discharge was performed in a voltage range of 0.01–3.0 V on a Land CT 2001A battery testing system (Land, China) at room temperature. The entire cell was assembled using KPB as cathode and NCNFs as anode, with the same separator and electrolyte. The cathode was fabricated in the same way as anode on an aluminum foil. The cathode-to-anode mass loading ratio was 4:1 and 2:1 in the anode-limited and cathode-limited configurations, respectively. Galvanostatic charge/discharge was performed in a voltage range of 2.0–4.2 V.

**DFT calculations**. All calculations were performed with Vienna ab initio simulation package (VASP)[65]. Generalized gradient approximation (GGA) with the function of Perdew–Burke–Ernzerhof (PBE)[66] was employed to describe the

electron interaction energy of exchange correlation. Grimme's semi-empirical DFT–D3[67] scheme was used in the computations to produce a better description of the interaction in a long range. In all calculations, the plane wave cutoff was set to 450 eV. The Hellmann–Feynman forces convergence criterion on the atoms was set to be lower than 0.02 eV Å$^{-1}$ during geometrical optimization. Tolerance of self-consistency was achieved at least 0.01 meV in the total energy. The Brillouin zone was sampled by Monkhorst-Pack[68] method, with Gamma centered to $3 \times 3 \times 1$ for a single-layer model and $3 \times 3 \times 2$ for a multiple-layer model. A vacuum layer of 14 Å was built to prevent interactions between the two repeated layers.

**Date availability**. The data that support the findings of this study are available from the corresponding authors on reasonable request.

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

### Acknowledgements

This work was financially supported by the European Research Council (ThreeDsurface, 240144, and HiNaPc, 737616), BMBF (ZIK–3DNanoDevice, 03Z1MN11), German Research Foundation (DFG: LE 2249_4–1), National Natural Science Foundation of China (21577086 and 41430644), the Shanghai Thousand Talent Plan, and Program for Changjiang Scholars and Innovative Research Team in University (IRT13078).

### Author contributions

Y.X. and C.L.Z. contributed equally to this work. Y.X., C.L.Z., and Y.L. conceived the idea, designed research plan, and wrote the paper. Y.X. and C.L.Z. carried out experiments and analyzed data. M.Z. and Q.F. contributed to the kinetics analysis and were involved in discussion on the electrochemical performance. C.X.Z. contributed to the theoretical simulations. Y.L., M.H.W., and Y.X. supervised the whole project. All the authors read the paper and made comments.

### Additional information

**Competing interests:** The authors declare no competing interests.

