## [Peer Review File · Nature Communications]

Reviewers' comments:

Reviewer #1 (Remarks to the Author):

This paper presents highly N-doped carbon fibers as novel anode materials in PIBs. The subject of the work is timely and of great interest to a broad audience of materials scientists, chemists and applied physicists. The work shows the outstanding electrochemical performance of the proposed electrode in terms of both specific capacity as well as charging rate and provides exhaustive experiments and calculations to determine the underlying mechanisms for the observed properties. I enjoyed the discussions presented in the paper which are based in robust experiments as well as DFT calculations. Therefore, I believe that this work fulfills the requirements for publication in Nature Communications.

Reviewer #2 (Remarks to the Author):

REVIEW: Highly nitrogen-doped carbon nanofibers as potassium-ion battery anodes with superior rate capability and cyclability.

The authors demonstrate a high-rate and extended cyclability soft carbon K anode, using a polypyrrole (PPy) nanofibers as a precursor. The carbon fibers exhibit very good reversible capacity and exceptional cyclability. Overall this is a great paper, well executed and with a solid combination of experiment and theory. The field of PIB anodes is young enough to make this work novel, and to make it fruitful for follow up studies by others. Few points where the manuscript could be improved:

1) Too many digits in the data, the potentiostats or BET systems are never that accurate and there are numerous small sources of error: Reported values of 248.1 mAh g⁻¹ should really be 248, CE = 49.46% should be CE = 49%, 90.2% pseudocapacitive contribution should really be 90%, etc. Having too many digits sends the wrong message to the reader about the data being over interpreted.

2) Cycling lifetime. The materials cycle well in part because the rates are fairly high and all the charge storage action is on or near the surface. Slower charge rates, which would give more bulk storage contribution, would certainly do more damage to the electrodes. This should be at least recognized and pointed out in a few sentences.

3) Strictly speaking the material is not pseudocapacitive. Oxides like MnO₂ are true pseudocapacitors in a sense that their galvanostatic charge – discharge profiles are nearly perfect triangles ($\Delta V/\Delta Q = \text{constant}$), while their CV curves are “box-like” without distinct redox peaks [Journal of The Electrochemical Society. 2015 Jan 1;162(5):A5185-9.] . This actually does not say anything about the charge – discharge mechanism. For instance for MnO₂ in an aqueous acidic electrolyte the “box-like” shape is due to the large number of oxidation states and intermediate MnO_x structures.

Really what is observed here is a surface – redox dominated process, where K binds with various surface and near surface heteroatom moieties and graphene defects. Since there is a distribution of adsorption site energies, the voltage vs. capacity profile winds up having some irregular slope. Similar behavior is observed with Na and Li non-graphitic soft and hard carbons at both high and low rates [Energy & Environmental Science. 2016;9(1):102-6; Journal of Materials Chemistry A. 2014;2(46):19685-95; Journal of Materials Chemistry A. 2016;4(14):5149-5; Electrochimica Acta. 2015 Sep 10;176:533-41; Nano Energy. 2015 Oct 31;17:43-51.].

Moreover the linear dependence of maximum current (max reaction rate) on time does not signal pseudocapacitance per se. It just indicates that charge storage in the carbons is reaction controlled,

rather than diffusion controlled. In standard electrochem terms it indicates activation polarization rather than diffusion polarization, without actually giving any specifics about the reaction mechanism. A modified discussion is recommended.

4) Coulombic efficiency and early cycling capacity loss. Very little is said about this important aspect. It is expected that for reversible binding at carbon defects and heteroatom functional groups, the early CE loss will be substantial. It is also expected that it will decrease with heat treatment temperature as defects/heteroatoms are eliminated. A separate discussion of this is recommended. The references above contain some points to consider in the regard to early cycling CE loss.

Response letter for the manuscript entitled “Highly nitrogen-doped carbon nanofibers as potassium-ion battery anodes with superior rate capability and cyclability” (NCOMMS-17-26467A-Z)

We sincerely appreciate the detailed and constructive comments put forth by the reviewers and now have revised the manuscript accordingly, with all the reviewers’ concerns being addressed. This letter details our point-by-point response to the reviewers’ comments with the analysis performed per their input. All the revisions have been highlighted in yellow in the revised manuscript.

Reviewer #1

This paper presents highly N-doped carbon fibers as novel anode materials in PIBs. The subject of the work is timely and of great interest to a broad audience of materials scientists, chemists and applied physicists. The work shows the outstanding electrochemical performance of the proposed electrode in terms of both specific capacity as well as charging rate and provides exhaustive experiments and calculations to determine the underlying mechanisms for the observed properties. I enjoyed the discussions presented in the paper which are based in robust experiments as well as DFT calculations. Therefore, I believe that this work fulfills the requirements for publication in Nature Communications.

Response: We highly appreciate the reviewer’s positive comments.

Reviewer #2

The authors demonstrate a high-rate and extended cyclability soft carbon K anode, using a polypyrrole (PPy) nanofibers as a precursor. The carbon fibers exhibit very good reversible capacity and exceptional cyclability. Overall this is a great paper, well executed and with a solid combination of experiment and theory. The field of PIB anodes is young enough to make this work novel, and to make it fruitful for follow up studies by others. Few points where the manuscript could be improved

Response: We highly appreciate the reviewer’s positive comments. We have revised our manuscript according to the points raised by the reviewer and improved the quality of our manuscript.

Comment 1: *Too many digits in the data, the potentiostats or BET systems are never that accurate and there are numerous small sources of error: Reported values of 248.1 mAh g⁻¹ should really be 248, CE = 49.46% should be CE = 49%, 90.2% pseudocapacitive contribution should really be 90%, etc. Having too many digits sends the wrong message to the reader about the data being over interpreted.*

Response: We agree with the reviewer that too many digits send the wrong message to the reader about the data being over interpreted. We have reduced the digits of the data in the revised manuscript and supplementary information. The data include the empirical R values derived from the XRD patterns, the BET surface areas, the content of elements and N-dopants in the XPS results, the capacities and CEs of the cells, and the charge storage contribution from surface process. We thank the reviewer for his/her suggestions.

Comment 2: *Cycling lifetime. The materials cycle well in part because the rates are fairly high and all the charge storage action is on or near the surface. Slower charge rates, which would give more bulk storage contribution, would certainly do more damage to the electrodes. This should be at least recognized and pointed out in a few sentences.*

Response: We thank the reviewer for the suggestion. It indeed is necessary to point out that the capacity retention at high and low rates is related to the different charge storage sites in the doped electrodes. NCNF-650 exhibited capacity retention of 92% after 100 cycles at 25 mA g⁻¹, and the retention increases to 98% at cycle100 at 2 A g⁻¹. It shows that charge storage on or near the surface delivers better cyclability. We have added the related text on page 11 of the revised manuscript:

“The long-term operation of the electrode without much degradation can be attributed to the charge storage action that is mainly on or near the surface of the nanofibers at high rates. It can be evidenced by a dominating charge storage contribution from the surface process, which will be discussed in the next section. Cycling at low rates would give more charge storage contribution in bulk and certainly do more damage to the electrode, as can be seen that NCNF-650 showed lower capacity retention after 100 cycles at 25 mA g⁻¹ comparing with those at high rates.”

Comment 3: *Strictly speaking the material is not pseudocapacitive. Oxides like MnO₂ are true pseudocapacitors in a sense that their galvanostatic charge – discharge profiles are nearly perfect triangles ($\Delta V/\Delta Q = \text{constant}$), while their CV curves are “box-like” without distinct redox*

peaks [Journal of The Electrochemical Society. 2015 Jan 1;162(5):A5185-9.] . This actually does not say anything about the charge – discharge mechanism. For instance for MnO₂ in an aqueous acidic electrolyte the “box-like” shape is due to the large number of oxidation states and intermediate MnO_x structures.

Really what is observed here is a surface – redox dominated process, where K binds with various surface and near surface heteroatom moieties and graphene defects. Since there is a distribution of adsorption site energies, the voltage vs. capacity profile winds up having some irregular slope. Similar behavior is observed with Na and Li non-graphitic soft and hard carbons at both high and low rates [Energy & Environmental Science. 2016;9(1):102-6; Journal of Materials Chemistry A. 2014;2(46):19685-95; Journal of Materials Chemistry A. 2016;4(14):5149-5; Electrochimica Acta. 2015 Sep 10;176:533-41; Nano Energy. 2015 Oct 31;17:43-51.].

Moreover the linear dependence of maximum current (max reaction rate) on time does not signal pseudocapacitance per se. It just indicates that charge storage in the carbons is reaction controlled, rather than diffusion controlled. In standard electrochem terms it indicates activation polarization rather than diffusion polarization, without actually giving any specifics about the reaction mechanism. A modified discussion is recommended.

Response: We are very much appreciative of the reviewer’s comments. The comments are insightful and bring out a key point of our work. First, we apologize for the inaccurate use of the term “pseudocapacitive.” We fully agree with the reviewer that what is observed in our work is essentially a surface-dominated process, rather than a “pseudocapacitive contribution”. The charge-discharge profiles do not reflect the characteristic of $\Delta V/\Delta Q$ being constant; neither do the CV curves show the “box-like” shape. Second, we have read the literature suggested by the reviewer, together with other related ones. The literature is very helpful for us to understand the nature of “being pseudocapacitive” and modify the discussion in the manuscript. The essence of the term “pseudocapacitive” is to exhibit a linear dependence of the charge stored with the width of the potential window, but where charge storage originates from different reaction mechanism.^[1] The literature also helps to clarify the misuse of the term in many previous papers, some of which were read during the preparation of this manuscript. Accordingly, we have modified the discussion in the revised manuscript and cited the suggested literature.

Again we thank the reviewer for his/her insightful and helpful comments and suggestions. The comments and suggestions help us to greatly improve the quality of the manuscript, and could benefit our future works.

Comment 4: *Coulombic efficiency and early cycling capacity loss. Very little is said about this important aspect. It is expected that for reversible binding at carbon defects and heteroatom functional groups, the early CE loss will be substantial. It is also expected that it will decrease with heat treatment temperature as defects/heteroatoms are eliminated. A separate discussion of this is recommended. The references above contain some points to consider in the regard to early cycling CE loss.*

Response: We highly appreciate the reviewer's valuable comments and apologize for not being able to interpret the important aspect of coulombic efficiency and early cycling capacity loss. In our work, the initial coulombic efficiencies (ICEs) are 49%, 38% and 36% for NCNF-650, NCNF-950 and NCNF-1100, respectively. The ICE loss is indeed substantial as the reviewer expected, and it does not decrease with elevating the heat treatment temperature. On one hand, we fully agree that the ICE loss should be expected to decrease with elevating heat treatment temperature as carbon defects/heteroatoms are eliminated. On the other hand, we think the different trend in our case could be attributed to the collective effects from carbon defects/heteroatoms, decomposition of electrolyte (i.e., formation of solid electrolyte interface (SEI)), and irreversible K-intercalation. We would like to explain it from the following aspects.

- (1) The formation of SEI has been considered as a major contributor to ICE loss for rechargeable ion-batteries, which is associated with high surface area of the nanostructured electrode materials. In the paper that is suggested by the reviewer, the non-SEI related CE loss is studied in the carbon materials with a very low surface area ($14.5 \text{ m}^2 \text{ g}^{-1}$).^[2] It disqualifies SEI from having any substantial role in the CE loss. In our case, the NCNFs have a surface area of $100\text{-}110 \text{ m}^2 \text{ g}^{-1}$. Although the surface area is not considered to be significantly high comparing with (meso)porous carbon materials, it could qualify SEI formation as one of the contributors to the ICE loss. Additionally, it has been reported in lithium-ion batteries that doping of nitrogen can, to some extent, suppress the electrolyte decomposition and SEI formation.^[3,4] We think NCNF-650 with the highest ICE among the NCNFs might benefit from its high nitrogen doping level in this perspective.
- (2) Irreversible K-intercalation could be another contributor to the ICE loss. Studies of K-intercalation in graphite have shown that major intercalation occurs at the voltage below 0.25 V (vs. K/K⁺).^[5-7] As seen in Fig. S8, NCNF-650 exhibits lowest capacity below 0.25 V among the three NCNFs, which indicates that there is less K intercalating between graphene

layers in NCNF-650 than in NCNF-950 and NCNF-1100. This is in accordance with the lower graphitization degree and surface-dominated charge storage contribution in NCNF-650. According to the center position of (002) peaks in the XRD patterns, the average graphene interlayer spacing are calculated to be 3.58, 3.47, and 3.44 Å for NCNF-650, NCNF-950, and NCNF-1100, respectively. On the basis of the less K-intercalation in NCNF-650 and its larger graphene interlayer spacing than NCNF-950 and NCNF-1100, we speculate that fewer K-intercalation is irreversible in NCNF-650, which thus contributes to its higher ICE over the other two NCNFs, even though K-intercalation does not appear to be a major contributor to the overall capacity.

We have conducted a literature survey of nitrogen-doped carbon materials (**Table R1**), and it shows that the relationship between heteroatoms/carbon defects and ICE loss is widely varied from carbon to carbon, depending on synthetic routes, doping method, graphitization degree, surface area, morphology, etc. Although the results in the table serve just as references to the nitrogen-doped carbon materials, and do not indicate a specific changing trend of ICE upon nitrogen content, it is safe to say that the relationship between heteroatoms/carbon defects and ICE loss can be influenced by collective effects.

Table R1. Summary of the previously reported nitrogen-doped carbon materials in lithium- and sodium-ion batteries (LIBs and SIBs)

Materials	Sample labels	N content [#] (at% or wt%)	ICE (%)	Current density (A g ⁻¹)	Reference/ Battery type
N-doped carbon nanofibers (CNFs)	CNFs	11.3	52.8	0.05	[8]/SIBs
	10U-CNFs	12.1	55.7		
	30U-CNFs	18.7	64.8		
N and O functionalized carbon (NOFC)	NOFC-650	10.2	52	0.1	[9]/SIBs
	NOFC-800	4.7	43		
	NOFC-950	1.9	40		
N-doped porous carbon (NPC)	NPC-1	11.5	59.4	0.1	[10]/LIBs
	NPC-2	9.8	62.8		
	NPC-3	8.4	62.5		

N-rich mesoporous carbon (NMC)	NMC-700	15.4	54	0.1 (mA cm ⁻²)	[11]/LIBs
	NMC-750	15.1	54		
	NMC-800	9.7	46		
	NMC-850	5.6	41		
	NMC-900	2.8	29		
Protein derived mesoporous carbon (PMC)	PMC-650	6.5	55	0.1	[12]/LIBs
	PMC-750	4.2	65		
	PMC-850	3.0	60		
N-doped carbon nanofibers (N-CNFs)	N-CNF-700	16.3	83.3	0.05	[13]/SIBs
	N-CNF-800	11.2	64.4		
	N-CNF-900	10.7	38.8		
Seaweed-like porous carbon (SPC)	SPC-3-600	8.2	59.8*	0.05	[14]/LIBs
	SPC-3-700	4.3	59.2*		
	SPC-3-800	1.8	58.1*		
N-doped carbon microspheres (NCSs)	NCSs-500	3.8	38.8*	0.05	[15]/SIBs
	NCSs-700	2.9	36.2*		
	NCSs-900	2.7	36.4*		
N-doped carbon nanosheets (PPyCs)	PPyC-400	12.4	31.3*	0.1	[16]/SIBs
	PPyC-600	13.3	31.7*		
	PPyC-800	2.4	37.0*		
N-doped carbon Microshperes (NCSs)	NCS-500	7.1	46.4*	0.05	[17]/LIBs
	NCS-700	7.3	54.0*		
	NCS-900	4.7	54.4*		

Numbers in italic type are the sum content of pyrrolic and pyridinic N.

* Estimated according to the figures.

In summary, the NCNFs in our work show substantial early CE loss, and the ICE decreases with elevating the heat treatment temperature. We think the collective effects from carbon defects/heteroatoms, SEI formation, and irreversible K-intercalation is responsible for the obtained results. Once again we deeply thank the reviewer for the highly constructive suggestion that is very helpful for us to improve the quality of our manuscript, and hope the reviewer would agree on our explanations. As recommended by the reviewer, we have added a separation discussion on the early CE loss on page 9 and 10 of the revised manuscript.

References

- [1] T. Brousse, D. Bélanger, J. W. Long, *J. Electrochem. Soc.* **2015**, *162*, A5185.
- [2] E. M. Lotfabad, P. Kalisvaart, A. Kohandehghan, D. Karpuzov, D. Mitlin, *J. Mater. Chem. A* **2014**, *2*, 19685.

- [3] Z. Wu, W. Ren, L. Xu, F. Li, H. Cheng, *ACS Nano*, **2011**, *5*, 5463.
- [4] T. Hu, X. Sun, H. Sun, G. Xin, D. Shao, C. Liu, J. Lian, *Phys. Chem. Chem. Phys.* **2014**, *16*, 1060.
- [5] Z. Jian, W. Luo, X. Ji, *J. Am. Chem. Soc.* **2015**, *137*, 11566.
- [6] W. Luo, J. Wan, B. Ozdemir, W. Bao, Y. Chen, J. Dai, H. Lin, Y. Xu, F. Gu, V. Barone, *Nano Lett.* **2015**, *15*, 7671.
- [7] J. Zhao, X. Zou, Y. Zhu, Y. Xu, C. Wang, *Adv. Funct. Mater.* **2016**, *26*, 8103.
- [8] C. Chen, Y. Lu, Y. Ge, J. Zhu, H. Jiang, Y. Li, Y. Hu, X. Zhang, *Energy Technology* **2016**, *4*, 1440.
- [9] J. Ding, Z. Li, K. Cui, S. Boyer, D. Karpuzov, D. Mitlin, *Nano Energy* **2016**, *23*, 129.
- [10] X. Zhang, G. Zhu, M. Wang, J. Li, T. Lu, L. Pan, *Carbon* **2017**, *116*, 686.
- [11] Y. Mao, H. Duan, B. Xu, L. Zhang, Y. Hu, C. Zhao, Z. Wang, L. Chen, Y. Yang, *Energy Environ. Sci.* **2012**, *5*, 7950.
- [12] Z. Li, Z. Xu, X. Tan, H. Wang, C. M. B. Holt, T. Stephenson, B. C. Olsen, D. Mitlin, *Energy Environ. Sci.* **2013**, *6*, 871.
- [13] J. Zhu, C. Chen, Y. Lu, Y. Ge, H. Jiang, K. Fu, X. Zhang, *Carbon* **2015**, *94*, 189.
- [14] X. Zhou, J. Tang, J. Yang, J. Xie, B. Huang, *J. Mater. Chem. A* **2013**, *1*, 5037.
- [15] D. Yang, C. Yu, X. Zhang, W. Qin, T. Lu, B. Hu, H. Li, L. Pan, *Electrochimica Acta* **2016**, *191*, 385.
- [16] Z. Luo, J. Zhou, X. Cao, S. Liu, Y. Cai, L. Wang, A. Pana, S. Liang, *Carbon* **2017**, *122*, 82.
- [17] T. Chen, L. Pan, T. A. J. Loh, D. H. C. Chua, Y. Yao, Q. Chen, D. Li, W. Qin, Z. Sun, *Dalton Trans.* **2014**, *43*, 14931.

Reviewers' comments:

Reviewer #2 (Remarks to the Author):

A fine response, acceptance recommended.

Reviewer #3 (Remarks to the Author):

The paper reported nitrogen-doped carbon nanofibers as anode for potassium ion batteries. Various carbon anodes have been reported as anode for K ion batteries in recent years, compared to the published work, this work does not show anything significantly new. The novelty of this paper is rather low in terms of both material reported and its electrochemical performance. Therefore, I cannot recommend publication of this paper.

Point-by-point response to the referees' comments for the manuscript entitled "Highly nitrogen-doped carbon nanofibers as potassium-ion battery anodes with superior rate capability and cyclability" (NCOMMS-17-26467B)

We sincerely appreciate the comments from the reviewers and now have revised the manuscript accordingly, with all the reviewers' concerns being addressed. This letter details our point-by-point response to the reviewers' comments with the analysis performed per their input. All the revisions have been highlighted in yellow in the revised manuscript.

Reviewer #2

A fine response, acceptance recommended.

Response: We highly appreciate the reviewer's positive comments.

Reviewer #3

The paper reported nitrogen doped carbon nanofibers as anode for potassium ion batteries. Various carbon anodes have been reported as anode for K ion batteries in recent years, compared to the published work, this work does not show anything significantly new. The novelty of this paper is rather low in terms of both material reported and its electrochemical performance. Therefore, I cannot recommend publication of this paper.

Response: We thank the reviewer for reading through our manuscript. However, we disagree with the reviewer's comment, as he/she may not have a comprehensive overview of the development of carbon anodes in potassium-ion batteries (PIBS) and made the comment solely based on the form of the carbon anodes and their performance, whereas overlooked the importance and potential influence of our manuscript. We believe that our manuscript is timely, and presents the most comprehensive and systematic work of carbon anodes in PIBs with the among-the-very-few best performance so far. We would like to detail our response point-to-point as following.

(1) "Various carbon anodes have been reported as anode for K ion batteries in recent years, compared to the published work, this work does not show anything significantly new."

The development of carbon anodes of PIBs is still in its infancy. To the best of our knowledge, there have been just more than ten publications of carbon anodes in PIBs so far, and those of doped carbon anodes are even less. Carbon materials have a great variety of structures (*e.g.*, morphology, porosity, size, composition, *etc.*), thus it hardly can say there have been "various"

carbon anodes, specially doped ones, in PIBs with that limited amount of publications, not even mentioning that the reported carbon structures are only the tip of the iceberg and the reported performance is different from case to case. Taking sodium-ion batteries (SIBs) as an example, there have been nearly 400 publications of carbon materials since 2015 according to the Web of Science. A comprehensive and systematic work like ours (will be discussed in the following) is much needed at this point, in order to push forward the development of PIBs and hopefully, intrigue more attentions to doped carbon anodes. Therefore, we believe that our manuscript is timely and among very few reports of doped carbon anodes, as commented by **Reviewer #1** that “The subject of the work is timely and of great interest to a broad audience of materials scientists, chemists and applied physicists” and **Reviewer #2** that “The field of PIB anodes is young enough to make this work novel, and to make it fruitful for follow up studies by others”.

(2) *“The novelty of this paper is rather low in terms of both material reported and its electrochemical performance.”*

We think this comment overlooks the underlying mechanism interpreted in our manuscript, and thus overlooks the novelty and potential influence of the manuscript.

- (i) Our manuscript presents **the full-cell performance of the highly nitrogen-doped nanofibers** (NCNFs). Full-cell performance is indispensable when demonstrating the practical application of carbon materials. As seen from the industry of lithium- and sodium-ion batteries, the commercialization of these batteries always starts from and relies on the successful utilization of carbon materials in full-cells. In our manuscript, the full-cells were tested in both anode- and cathode-limited configurations, and the NCNFs displayed the performance as well as they did in the half-cells. The results fully demonstrate the potential of the NCNFs toward the future commercialization of PIBs.
- (ii) Our manuscript **quantitatively interprets the correlation between the type and doping level of nitrogen dopant and the PIB performance of NCNFs**, using the experimental and theoretical analysis. The interpretation is critical to fully address the importance of doping carbon materials with nitrogen and more importantly, the necessity of controlling the type and doping level of each different nitrogen dopant. We believe that an investigation of carbon anodes in PIBs should be far more than simply developing a synthetic and/or doping method, and should provide the understanding of the doping effect on the electrochemical performance. As a result, the understanding provided in our manuscript could shed lights on the material design for a wider range of electrochemical energy devices where surface dominated charge storage plays an importance role.

- (iii) Our manuscript **highlights the importance of soft carbon anodes for K-storage**, which is the adjustable degree of graphitization and concentration of foreign dopants. This is in contrast with the case of hard carbons, where there is a very limited flexibility of adjusting the structure of the carbons at an atomic level. For some cases, the doping level is completely not adjustable if using the hard carbons derived from the carbonization of biomass. We believe it is the one of the most important merits of our manuscript to choose soft carbon as the target material and emphasize the advantages of soft carbons as PIB anodes.
- (iv) Additionally, the PIB performance presented in our manuscript, **which is among the very few best results, is obtained by the sample (NCNF-650) that has the smallest surface area among all the doped carbons reported so far.** It is also worth noting that the reported long-term cyclability at high rates is the best among all the carbons, both non-doped and doped carbons, in PIBs. This is, in turn, supporting the necessity of studying soft carbons that could maximize their advantages to store K-ions, i.e., adjustable degree of graphitization and concentration of foreign dopants.

In summary, we strongly believe that our manuscript is worth being published in a highly prestigious journal like *Nature Commun.*, in which the manuscript could be read by a wide research community, potentially benefit research on a large scale, and thus reach a wide readership.

During the review process of our manuscript, we noticed a recent publication (Energy Storage Materials, 2017, 8, 161) reporting a nitrogen-doped carbon anode with a similar PIB anode performance as ours. However, the reported anode was hard carbon whose nitrogen doping level was not presented and would have not been adjusted either. Full-cell performance was also missing. Thus, it is fundamentally different from our manuscript. We have cited this paper in the revised manuscript, and added a few sentences in the “Introduction” to describe the paper and emphasize the difference of it from our work. The related text is:

“N-doped hard carbon microspheres reported by Chen et al. exhibited high rate capability that was enabled by the surface-driven K-storage³⁵. The correlation of N-doping species to the K-storage performance was not presented due to the natural difficulty of controlling the N-doping level when dealing with hard carbons, and the full-cell demonstration was missing too. Both aspects are critical to utilize N-doped carbons to a maximum extent. Given the advantages of soft carbon over graphitic and hard carbon in PIBs²², it is of high importance to study N-doped soft carbon and correlation of N-doping species to K-storage performance.”

Reviewers' comments:

Reviewer #2 (Remarks to the Author):

A fine response, acceptance recommended.

Reviewer #3 (Remarks to the Author):

The paper reported nitrogen-doped carbon nanofibers as anode for potassium ion batteries. Various carbon anodes have been reported as anode for K ion batteries in recent years, compared to the published work, this work does not show anything significantly new. The novelty of this paper is rather low in terms of both material reported and its electrochemical performance. Therefore, I cannot recommend publication of this paper.